# Risk of COVID-19 re-infection and its predictors (CORES): protocol for a community-based longitudinal cohort study in Vellore, India

Ramya Madhavan,[1] Jackwin Sam Paul,[2] Sudhir Babji,[1] Dilesh Kumar,[1] Savit B Prabhu,[1] Harsha Kandi Pulleri,[3] Ravikiran Annadorai,[3] Sampreeth Ravi Gowda,[3] Jacob John,[2] Gagandeep Kang [1]

RM and JSP contributed equally.

RM and JSP are joint first authors.

For numbered affiliations see end of article.

**Correspondence to**
Professor Gagandeep Kang; gkang@cmcvellore.ac.in

## ABSTRACT

**Introduction** The incidence of SARS-CoV-2 re-infection has not been widely evaluated in low-income and middle-income countries. Understanding immune responses elicited by SARS-CoV-2 natural infection and factors that lead to re-infection in a community setting is important for public health policy. We aim to investigate the risk of primary infection and re-infection among those without and with evidence of prior infection as defined by the presence of antibodies to SARS-CoV-2 spike protein.

**Methods and analysis** A baseline seroprevalence survey will test for SARS-CoV-2 antibodies among healthy adults in Vellore, India. Based on an expected seropositivity rate of 50% in the general population, with annual attack rates of 12%, 6%, 4.8% and 4% among those unvaccinated and seronegative, vaccinated and seronegative, unvaccinated and seropositive, and vaccinated and seropositive, respectively, we will recruit 1200 adults who will be followed up for a total of 24 months. Weekly self-collected saliva samples will be tested by reverse transcription-PCR (RT-PCR) to detect SARS-CoV-2 infections, for a period of 1 year. For any person testing RT-PCR positive, blood samples will be collected within 2 days of RT-PCR positivity and on days 30 and 90 to assess the kinetics and longevity of the antibody responses, B cell memory and T cell memory post-infection. The data will be analysed to estimate seroprevalence at baseline and over time, the risk factors for infection, rates of primary infection and re-infection, and provide a comparison of the rates across groups based on infection and vaccination status.

**Ethics and dissemination** The study has been approved by the Institutional Review Board (IRB No: 13585) of Christian Medical College and Hospital, Vellore. The results of the study will be made available through journal publications and conference presentations.

**Trial registration number** Central Trial Registry of India: CTRI/2020/11/029438.

## STRENGTHS AND LIMITATIONS OF THIS STUDY

⇒ The use of saliva samples for SARS-CoV-2 surveillance will be an acceptable alternate as it is self-directed and non-invasive.
⇒ Weekly salivary reverse transcription-PCR will serve as surveillance for SARS-CoV-2 at the community level in Vellore.
⇒ The study involves analysis of both humoral and cellular immune responses in individuals with infections and re-infections.
⇒ The immunological profile following vaccine breakthrough infections will be studied in detail.
⇒ Though there is a good concordance of saliva and nasopharyngeal swab for SARS-CoV-2 surveillance, there could be some infections which may be missed with saliva sampling.

## INTRODUCTION

Immune responses to SARS-CoV-2 infection, vaccination and the immune correlates of protection are areas of active investigation.[1–4] A few studies have shown that the development, amount and kinetics of antibodies may correlate with the clinical outcome of SARS-CoV-2 infections.[5–7] The coordinated response between humoral and cellular immunity has been hypothesised to be protective.[8] From a public health perspective, it is crucial to understand the duration of protective immunity offered by natural infections and vaccination. The reported duration of protection following a natural infection is around 8 months to 1 year.[2–4] Re-infections from a different strain have been documented in persons who have recovered from a prior natural infection.[9 10] At the population level, the incidence of re-infection over a longer term of 1–2 years due to various variants of concern (VOCs) has not been evaluated, and this is also affected by vaccination. Preliminary studies suggest that antibodies persist for 7–9 months or more post-infection.[11 12] The rates of attrition of potential immune correlates like memory B and T cell responses, and the association of these humoral and cellular immune parameters with subsequent re-infections, particularly with VOCs, are

unknown. The duration of protective immunity to SARS-CoV-2 is being measured, but so far has largely been extrapolated from the data of phylogenetically related viruses. Antibody responses to SARS-CoV-1 persist for 2–3 years[13] and memory T cells persist for 11 years after infection.[14] In contrast, beta coronaviruses (β-CoV) that are phylogenetically close to SARS-CoV-2 are known to re-infect humans throughout life,[15] suggesting short-lasting protective immunity. Human-controlled infection models using common cold-associated β-CoV showed partial protection from antibodies that persist for 1 year.[16] These findings suggest that similar protective immune mechanisms could be operative in SARS-CoV-2 as well but need detailed characterisation in populations with known viral circulation. Further, uninfected individuals could harbour antibodies and memory T cells to other β-CoV.[17] Such cross-reactive T cell responses,[17] targeting several epitopes on the surface proteins of SARS-CoV-2, could potentially influence the course of infection or the clinical outcomes. The limited availability of data on SARS-CoV-2 infections in low-income and middle-income countries, where exposure to other coronaviruses may differ, warrants a detailed evaluation of cross-reactive T cell and antibody landscapes in primary infections and re-infection outcomes in the community.

This protocol describes a study to estimate the incidence of infection, re-infection and vaccine breakthrough infections in a community in India. The study would also determine the antibody profile, duration of antibody persistence as well the cellular immune responses following natural COVID-19 infection and re-infection.

### Objectives and expected outcomes

The CORES Study has the objectives and outcomes as described in box 1.

## METHODS AND ANALYSIS
### Study setting

Vellore is a tier 2 city in northern Tamil Nadu with a population of close to 500 000. It is divided into 4 zones and 60 administrative wards. The Vellore Health and Demographic Surveillance System (VHDSS), established by the Christian Medical College (CMC), monitors a population of 120 000 people across zones 3 and 4 of the city. This study area has a very high population density predominantly belonging to the economically poorer section, and is largely homogeneous, with daily wage earners being the largest subgroup of the population.

### Study design

The study will have three components: (1) serosurvey to estimate the seroprevalence of SARS-CoV-2 spike protein antibodies in the study area; (2) prospective weekly follow-up to estimate the infection and re-infection rates in a cohort of 1200 individuals; (3) intensive follow-up of incident SARS-CoV-2 infections (both symptomatic and asymptomatic) to characterise immunological and

---

> **Box 1    Objectives and outcomes of the CORES Study**
>
> **Objective 1: To estimate the seroprevalence of antibodies to SARS-CoV-2 spike protein in Vellore (May–October 2021)**
> Outcome:
> a. The proportion of individuals ≥18 years of age who are seropositive for antibodies to spike protein of SARS-CoV-2 in Vellore.
> b. Prevalence of seropositivity across clusters (wards).
>
> **Objective 2: To measure the incidence of SARS-CoV-2 infection in a cohort of individuals ≥18 years in Vellore (May 2021–October 2023)**
> Outcome:
> a. Incidence of SARS-CoV-2 infection among those without evidence of prior infection or vaccination.
> b. Incidence of SARS-CoV-2 infection among those with evidence of prior SARS-CoV-2 infection.
> c. Incidence of SARS-CoV-2 infection in those who have received at least one dose of COVID-19 vaccine at least 14 days prior to infection.
>
> **Objective 3: To track cellular and humoral immune correlates of COVID-19 infection, re-infection and clinically significant disease (May 2021–October 2023)**
> Outcome:
> a. Kinetics and longevity of antibody responses and immunological memory.
> b. Influence of baseline memory T and B levels (both SARS-CoV-2 specific as well as cross-reactive) on infection.

clinical features of infection in the cohort. The study flow is in figure 1.

### Study status and timeline

The cohort recruitment started on 19 May 2021 and was completed on 28 October 2021. The cohort will be followed up for a period of 2 years and data will be collected until October 2023.

### Patient and public involvement

No patients or members of the public were involved in the design or conduct of the study. We will report the data in peer-reviewed publications and share it with state health authorities. Participants will be provided with study results and interpretation at a public meeting at the end of the study.

### Inclusion and exclusion criteria

Inclusion criteria:
1. Age 18 years and above.
2. Permanent residents of the selected wards.
3. Only one member from each selected household will be enrolled.
4. Individuals with a history of clinical illness suggestive of COVID-19 or confirmed COVID-19 in the past, who are seropositive at baseline in the serosurvey (symptomatic seropositive).
5. Individuals seropositive at baseline, with no history of COVID-19 (asymptomatic seropositive).

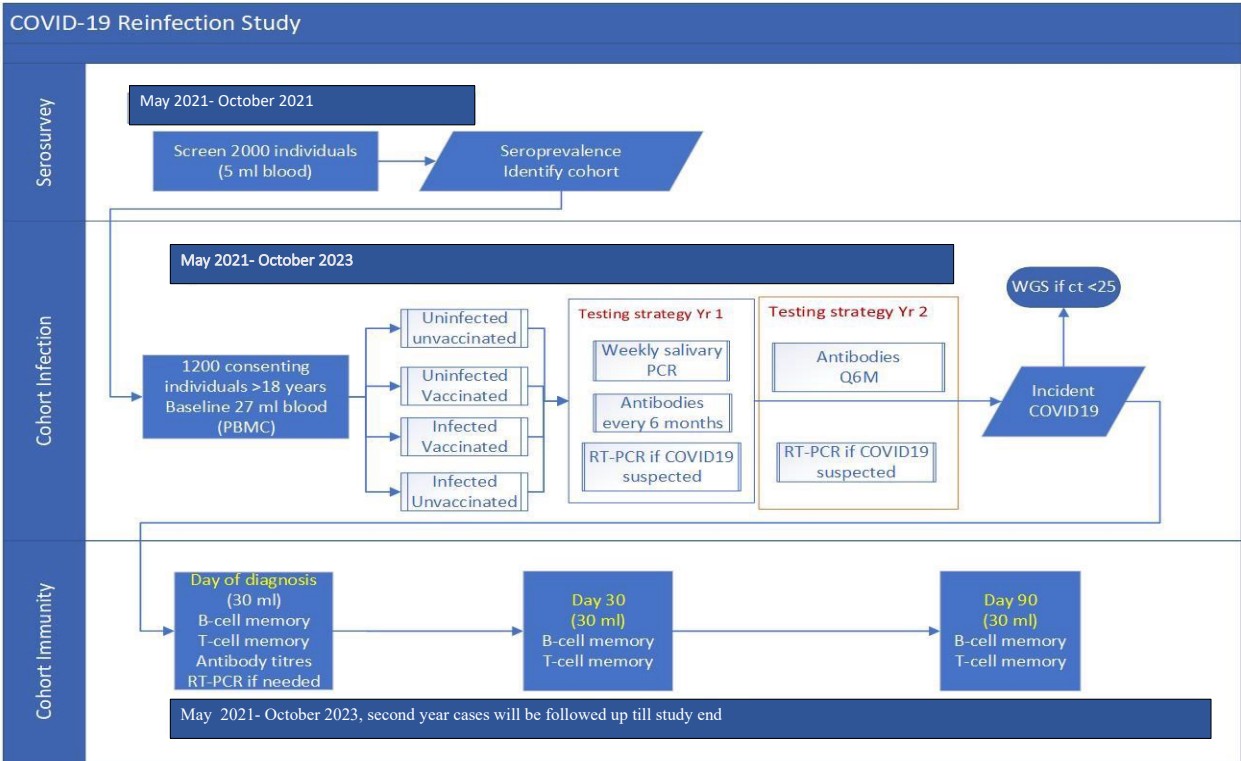

**Figure 1** CORES Study flow chart. PBMC, peripheral blood mononuclear cell; RT-PCR, reverse transcription-PCR, WGS-Whole Genome Sequencing, ct- cycle threshold, Q6M- Every 6 Month.

6. Individuals seronegative at baseline, stratified by the ward of residence.
   Exclusion criteria:
1. Participant refusal of consent.
2. Pregnant women and immunocompromised patients.
3. Participants not willing for follow-up until the end of the study.
4. Active cancers or bleeding disorders.

## Statistical considerations
### Assumptions
We make the following assumptions. The assumptions were based on the early findings of the Com-CoV study as there were no published data regarding vaccine efficacy and re-infections prior to the start of the study.[18]
1. Fifty per cent of the population will be seropositive at baseline.
2. Forty per cent will have received two doses of vaccine mid-way into the study.
3. The annual incidence of SARS-CoV-2 infection detected by the salivary PCR in those unvaccinated and have no detectable antibodies (unexposed) at baseline will be 12%.
4. The annual incidence of SARS-CoV-2 infection detected by the salivary PCR in those vaccinated and unexposed at baseline will be 6% (Vaccine Effectiveness (VE) 50% against infection).
5. The annual incidence of SARS-CoV-2 infection detected by the salivary PCR in those unvaccinated and who have antibodies (exposed) at baseline will be 4.8%.

6. The annual incidence of SARS-CoV-2 infection detected by the salivary PCR in those vaccinated and exposed at baseline will be 4%.

Based on these assumptions, for 90% power to detect a 5% difference in the rate of re-infection and primary infection in the cohort, a sample size of 1200 participants is proposed, after allowing for a 10% dropout rate.

## KEY DEFINITIONS
► Seropositive is defined as serum/plasma samples positive for IgG spike protein antibody to SARS-CoV-2 identified by LIAISON SARS-CoV-2 TrimericS IgG assay by Diasorin platform. The cut-off for seropositivity is more than or equal to 33.8 BAU/mL.
► Past asymptomatic infection refers to those who are seropositive (or documented reverse transcription-PCR (RT-PCR) positive >1 month in the past) but are neither antigen or RT-PCR positive at baseline assessment AND have had no symptoms of COVID-19.
► Recent asymptomatic infection refers to those who are seronegative AND are either RT-PCR or antigen positive AND have had no symptoms of COVID-19.
► Past symptomatic infection refers to those who are seropositive (or documented RT-PCR positive >1 month in the past) but are neither antigen or RT-PCR positive at assessment AND have had symptoms of COVID-19 in the past.

- ► Recent symptomatic infection refers to those who are seronegative AND are either RT-PCR or antigen positive at assessment AND have symptoms of COVID-19 within the past 1 month.
- ► Clinically significant disease refers to those who develop symptoms due to SARS-CoV-2 and require hospitalisation or intensive care unit admission.
- ► Re-positivity refers to those who test positive within 90 days of the first RT-PCR results with symptoms.
- ► Re-infection refers to those who test positive after 90 days of the first RT-PCR results with or without any symptoms.

## Study procedures

### Baseline serology screening

A baseline serosurvey, conducted on 2000 individuals in four urban clusters, is planned based on population proportionate to size. The participants who satisfy the inclusion criteria will be selected for the serosurvey from areas within the Vellore corporation limits after obtaining written informed consent. The inclusion and exclusion criteria are detailed in the earlier section. The baseline demographic information, along with details of any clinically relevant illness in the past 1 month, will be documented. History of confirmed COVID-19 or COVID-19-like illnesses during the period of the pandemic will also be documented. A peripheral blood sample (5 mL serum) will be collected.

### Establishment of the cohort

Based on the seroprevalence from the serosurvey, longitudinal follow-up will be initiated in the VHDSS area. A total of 1200 residents living in the densely populated wards of zones 3 and 4 of the Vellore corporation will be recruited for the longitudinal follow-up. Those subjects who agree to the specific terms of the longitudinal follow-up of 24 months will be recruited after informed consent. One member in the household will be selected using simple random sampling. Each study participant will be assigned a unique cohort ID. The final 1200 participants will be in any of the four groups based on their vaccination and infection status with no specific distribution across these four groups. The vaccination status will be obtained and recorded at the baseline and every 6 months for those who were unvaccinated at enrolment. Details of precautionary or booster doses will also be captured during the 6 monthly interview. The vaccination certificate would be verified for confirmation of details (date, type of vaccine, number of doses, etc). Upon recruitment, blood samples (15–30 mL) will be collected, processed and stored as per standard protocol. Peripheral blood mononuclear cells (PBMCs) will be isolated and stored to assess the baseline memory T cell and B cell profiles in the future.

### Intensive follow-up phase

The first year following recruitment of the cohort would be the intensive follow-up phase during which weekly follow-up visits and saliva sampling are planned.

An assigned field research assistant (FRA) will contact the study participant every week, either by telephonic or direct visit, and collect information regarding any COVID-19-like symptoms in the preceding week. The study participants will be trained to collect 2 ml of saliva in the universal sample container, early in the morning, on one designated day of the week. The participants will be asked to collect these samples as per the study protocol, prior to routine oral hygiene, and consumption of any food or drink. The samples will be collected by the FRAs and transported to the laboratory in vaccine carriers with ice packs to maintain a temperature of 4°C. The samples, once received in the laboratory, will be aliquoted in two vials. One vial will be retained at the Wellcome Trust Research Laboratory, Vellore. The other vial is sent to the National Centre for Biological Sciences, Bangalore (NCBS) for RT-PCR.

If an individual tests positive for SARS-CoV-2, the weekly salivary sample collection will be suspended for the next 90 days. The weekly contact, however, will be continued. During the weekly contact, if a subject develops symptoms, their samples will be collected. If they are RT-PCR positive within 90 days, it will be considered as re-positive. The study participants will be requested to inform the study team if they experience any clinically significant febrile or respiratory distress. Symptomatic individuals will be advised to visit CMC Hospital, Vellore and get tested by nasopharyngeal RT-PCR, as deemed necessary, after clinical examination. Clinical symptoms, response to treatment and details of treatment during hospitalisation or during home management would be recorded on the case report form (CRF) for every participant who is positive by RT-PCR.

### Follow-up phase: second year

During the second year of the study, weekly follow-up would be through telephonic interviews. Weekly salivary samples will not be collected, and home visits will be done only for subjects with symptoms. Any incident infection will be followed up for detailed immunological testing. Once every 6 months, a blood sample (5 mL) will be collected for assessing the serostatus of the participants to identify any infection that was missed through the RT-PCR screening. Sequencing will be done on all positive samples to identify the genetic sequence of the virus at NCBS, Bangalore and help us determine which VOC was responsible for the infections and re-infections. This will include samples classified as 're-positives'.

### Detailed follow-up of COVID-19 infections

All COVID-19 infections, including symptomatic and asymptomatic, will be followed up from the day of the positive report (day 0). Blood samples (30 mL) will be collected for PBMC isolation and storage within 24 hours of identification of positives on day 0, day 30 (+2 days) and day 90 (+7 days) post-infection.

## Sample collection

*Blood sample—serology*: 5 mL of peripheral blood will be collected (in serum tubes) from 2000 individuals during the baseline serosurvey, and once every 6 months from the 1200 study participants who are part of the longitudinal cohort.

*Salivary sample*: salivary samples will be self-collected, stored and transported to the NCBS laboratory, Bengaluru, as per the standard operating procedure, once a week during the first year of the study. The results will be uploaded into the secure data entry portal designed for the laboratory.

*Nasopharyngeal swab*: if any study participants report any clinically significant febrile illness or respiratory distress, they will be offered a medical consultation, and when necessary, a nasopharyngeal RT-PCR at CMC, Vellore or in any institute of their choice.

*Blood sample (for PBMC)*: 30 mL (minimum 15 mL) of blood will be collected (in 9 mL heparin tubes) after recruitment into the longitudinal study and for confirmed SARS-CoV-2 infections on day 0, day 30 and day 90. PBMCs will be separated by density gradient centrifugation method and cryopreserved in liquid nitrogen.

## Laboratory procedures

### Weekly salivary samples

Upon receipt and aliquoting, salivary samples will be pooled for testing on the same day. Ten microlitres of five samples each will be pooled in a single well of the PCR plate, and 6 µL of proteinase K of 50 µg/µL concentration will be added to each well. Subsequently, the plates will be sealed and heated at 95°C for 5 min in a dry thermal bath. After heat inactivation, the plates will be stored at −80°C. The pooled PCR plate and an aliquot of saliva will be transported on dry ice to NCBS. RT-PCR will be performed on the pooled samples targeting the N gene, E gene and RdRp gene of SARS-CoV-2. The limit of detection of the commercial kit that is used for testing is 100 copies/mL and the sensitivity of detection in saliva samples is around 94% compared with nasopharyngeal swab.[19] If any pool turns out to be positive, RT-PCR will be performed on individual samples. All positive samples will undergo sequencing.

### Blood samples

#### Serological assays

The plasma or serum sample collected at different time points will be tested for IgG antibody against spike protein using a high throughput automated platform (Diasorin LiaisonXL).

#### Immunophenotyping

Quantitation of SARS-CoV-2-specific T cells will be done by flow cytometric detection of cytokines and activation induced marker upregulation in T cells after stimulation with peptide pools. PBMC stimulation will be done using a 10-mer peptide pool for CD8 and 20-mer peptide-pools for CD4 T cells. Four peptide pools will be used, corresponding to the major proteins of SARS-CoV-2 (spike, envelope, membrane and nucleoprotein). For all the stimulation conditions, one well (vehicle-treated) will act as negative control. An additional well of cytomegalovirus (CMV)-peptide-stimulated control (a mix of 10-mer and 15-mer CMV peptides) will be kept as positive control for each sample. Baseline levels of cross-reactive T cells to non-SARS-CoV-2 human coronaviruses (hCoV) will be estimated using the same methodology, using peptide pools derived from hCoV strains. Memory B cells will be detected by flow cytometry after staining PBMCs with fluorophore-tagged viral proteins and memory B cell markers.

### Statistical analysis

Seroprevalence is estimated as a proportion and will be assumed to follow a binomial distribution. The incidence of infection within the cohort is expected to follow a Poisson distribution. We will permit repeated infections to be captured in analysis and account for the same in the analysis. A time-to-event analysis using Prentice, Williams and Peterson models comparing incidence in the exposed and unexposed cohorts will be performed. We will adjust for background infection rates in each cluster (ward) and covariates such as age, socioeconomic status, vaccination status, per-capita floor space and occupation class.

The statistical analysis plan will detail the estimation of seroprevalence, its risk factors, the incidence of primary and re-infection, and a comparison of these rates. Continuous variables will be described using mean (SD) and median (IQR) where necessary. Categorical data will be expressed as frequency (%). Incidence of infection and re-infection will be calculated per thousand person-years. HRs will be estimated to assess protection/risk conferred by vaccination and previous infection.

### Key comparisons in the study

We will make comparisons between:
- Incidence rates of infection overall and in seropositive and seronegative subgroups.
- Incidence rates of infection among the vaccinated individuals in the cohorts.
- Kinetics and longevity of memory B and T cells in infections occurring in the seropositive and seronegative cohort.
- Baseline cross-reactive T cells and antibodies to non-SARS-CoV-2 β-CoV between symptomatic infections versus asymptomatic infections versus uninfected individuals in the seronegative cohort.
- Baseline SARS-CoV-2-specific memory T and B cells and antibody levels between infected individuals versus uninfected individuals in the seropositive cohort.

### Data management plan

All the CRFs will be in the electronic format (REDCap), and the entry platform will be connected to the central database server. The data management system is

responsible for the periodical validation process and quality of the data. Any further correction in the database after the entry is 'saved' is accompanied by a duly completed 'data clarification form'. The electronic data management system tracks key study progress parameters on an access-restricted online dashboard. The weekly contact made by the FRAs will be independently validated by a field worker who calls 5% of all individuals who were contacted that week.

## Ethics and dissemination

The study has been approved by the Institutional Review Board (IRB No: 13585) of CMC, Vellore. The study will adhere to the principles that govern biomedical research involving human subjects as required in India. The Declaration of Helsinki will be followed to assure that the rights, integrity and confidentiality of study participants are protected, and that reported results are credible and accurate. The privacy and confidentiality of all information collected, including those derived from clinical specimens, will be ensured during and after the project. Individuals will not be identified in any reports or publications based on the study. All participant data will be computerised using password protection. The participants will be asked to provide written informed consent. The knowledge gained and the results will be made available through journal publications and conference presentations.

## DISCUSSION

To our knowledge, this study is the first to follow up a cohort in India, for a period of 2 years for COVID-19 infection and re-infection. In terms of surveillance of SARS-CoV-2 infection, though the nasopharyngeal swab has been the gold standard for diagnosis, the use of saliva samples will be an acceptable alternate by the study participants as it is self-directed, non-invasive and has a good concordance with the nasopharyngeal swab. The study aims to address several gaps in the current scientific evidence of SARS-CoV-2 infection and immunity. First, there are a limited number of studies that investigate the long-term follow-up of individuals for the rates of infection and re-infection in the community. Second, the study aims to look at the kinetics of IgG antibodies following infection. The cross-reactivity between SARS-CoV-2 and other hCoV will support better understanding of determinants of symptomatic infection. The T cell and B cell memory responses would help in understanding the kinetics and longevity of immune responses in seropositive and seronegative individuals and would help in decision-making with regard to booster vaccination. By studying the immunity and the risk of re-infection, we can potentially understand the factors that contribute to symptomatic COVID-19 infections. The study design will also allow the study of how the various VOCs contribute to re-infections. Large-scale vaccination had begun by the time enrolment had been completed. We anticipate that the majority of participants will be vaccinated at the end of the study and would have a hybrid immunity resulting from past infection and vaccine. In view of the 1-year intensive follow-up that requires weekly samples, we have planned to use salivary RT-PCR and only symptomatic individuals will receive nasopharyngeal swab for RT-PCR.

To conclude, CORES will help in estimating the re-infection rates, detailed immunogenicity among the COVID-19-positive individuals, establish the antibody kinetics and characterise the breakthrough infections among the vaccinated individuals in the community.

**Author affiliations**
[1]The Wellcome Trust Research Laboratory, Division of Gastrointestinal sciences, Christian Medical College and Hospital, Vellore, Tamil Nadu, India
[2]Department of Community Health and Development, Christian Medical College and Hospital, Vellore, Tamil Nadu, India
[3]COVID-19 Testing and Sequencing lab, Institute for stem cell science and Regenerative Medicine (inStem), Bangalore, Karnataka, India

**Acknowledgements** We acknowledge the participants who are willing to participate in the current study and help us in understanding the current knowledge gaps in COVID-19 infection and re-infections. We also thank the National Centre for Biological Sciences, Bangalore for helping us in processing our weekly saliva samples.

**Contributors** The study design and concept were conceived by JJ and GK. RM will conduct the study as part of her PhD under the supervision of GK, SB, SBP and JJ. JSP and JJ designed the process evaluation and wrote the statistical analysis plan; and JJ, DK and JSP organised data management and will oversee field operations. HKP, RA and SRG performed the RT-PCR of the weekly saliva samples. All authors provided edits and critiqued the manuscript for the scientific content. All authors read and approved the final version of the manuscript.

**Funding** This study is funded by the Bill and Melinda Gates Foundation (INV-024915).

**ORCID iD**
Gagandeep Kang http://orcid.org/0000-0002-3656-564X

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
