## [Reviewer comments · BMJ Open]

ARTICLE DETAILS

TITLE (PROVISIONAL)	Risk of COVID-19 re-infection and its predictors (CORES): protocol for a community based longitudinal cohort study in Vellore, India
AUTHORS	Madhavan, Ramya; Paul, Jackwin; Babji, Sudhir; Kumar, Dilesh; Prabhu, Savit; Pulleri, Harsha; Annadorai, Ravikiran; Gowda, Sampreeth; John, Jacob; Kang, Gagandeep

VERSION 1 – REVIEW

REVIEWER	West, Jack University Hospitals Dorset NHS Foundation Trust
REVIEW RETURNED	27-Jan-2022

GENERAL COMMENTS	A succinct design for a timely study. Longitudinal analysis will likely prove key to returning the world to normality. COVID-19 in LMICs is an under appreciated area that warrants investigation. Salivary samples are a pragmatic approach to enhancing compliance.
---

REVIEWER	Davido, Benjamin APHP, Maladies Infectieuses, Hôpital Universitaire Raymond-Poincaré
REVIEW RETURNED	10-Mar-2022

GENERAL COMMENTS	I have a major concern regarding the interest of publishing such study protocol. First, as far as I understand, the protocol already started mid-July 2021 in India, just after the wave of Delta VOC, where there was a very low number of cases (including primary infection). The reality of the situation is that the current Omicron Wave with BA2 achieved a number of substantial reinfection and already answered to that question. Second, my major concerns is that the article does not use the word VOC throughout all the manuscript, while in my opinion this is the main criteria of interest when studying the risk of reinfection. Moreover, the references are old and some new publications are lacking, therefore some statement such as line 123 "it is crucial to understand the duration of protective immunity" is outdated and referring to 2020 , just after you say they do need a one-year hindsight to analyse the risk of reinfection. Thus you might need recent publication because some studied that point. Third, the discussion is lacking the interest of those results, for instance whether or not immunity and risk of further reinfection does protect against severe form of covid and re-admission etc; and whether or not VOC does play a role. I'm sorry but, if the study has been achieved then I would rather suggest to publish the findings directly.
---

REVIEWER	Herman, Bumi Chulalongkorn University, Public Health
REVIEW RETURNED	10-Mar-2022

GENERAL COMMENTS	To the authors, these are some minors concern that could be addressed prior to study.  1. When is the anticipated date of study? 2. What is the consideration of excluding pregnant women and immunocompromised individuals in serosurvey? 3. How the researcher select one family member in the household? There should be a strict criteria regarding permanent residents, at least already residing in the community for three month etc. 4. Since this is a two-year prospective cohort, how the research accommodate booster vaccination into the survey? 5. It seems that there are 4 groups of participants in this cohort, Are there any articles or evidence that support the percentage of infection in these groups as assumptions for sample size estimation? How is the true situation of entire India prior to this study? Statement number 3, annual incidence in unvaccinated and unexposed was estimated at 12% whereas in vaccinated and unexposed was 6%. A study in US demonstrates that incidence rate of unvaccinated was 3-4 times higher(1) Is it too low to say that the vaccine efficacy is only 50%? 6. Line 255, supposed that symptoms suggesting COVID-19 occurs less than 90 days and tested positive in RT-PCR, will these case counted as reinfection? It is important to clearly define the term reinfection, and repositivity as these terms are different. 7. Line 292, is it possible to have other target gene such as ORF1b or S gene in RT-PCR as well? What is the level of detection of the proposed machine (LOD) and how sensitive the machine in treating saliva sample? 8. If feasible, will the researcher also collected the data regarding COVID medication received by the participants if tested positive? This could answer some questions of whether antivirus or other COVID medications could prevent re-infection. 9. As for defining seropositive status, kindly cite the minimum level of IgG that could be detected as seropositive according to the selected modalities 10. It is important to provide detailed statistical analysis plan, although the plan to use Prentice, Williams and Peterson models is appropriate for recurrent event. Reference 1. Tabak YP, Sun X, Brennan TA, Chaguturu SK. Incidence and Estimated Vaccine Effectiveness Against Symptomatic SARS-CoV-2 Infection Among Persons Tested in US Retail Locations, May 1 to August 7, 2021. JAMA Network Open. 2021;4(12):e2143346-e.
--

REVIEWER	Groome, Michelle National Institute for Communicable Diseases
REVIEW RETURNED	23-Mar-2022

GENERAL COMMENTS	The study fulfils the criteria for publication in the journal. This is well-written manuscript describing a cohort study being conducted in Vellore, India. This study will provide valuable information on SARS-CoV-2 infections and reinfections as well as a better understanding of the immune responses elicited by SARS-CoV-2 infection in a low-middle country setting. Strengths of the study include the long follow up period, intensive follow up of participants (weekly for the first year), good sample size, ability to look at humoral and cellular
--

	responses and the sequencing of positive samples, I wish you all the best for the important study. In general, the methods are clear. There are a few minor grammatical errors and some points for clarification as follows:  1. Objectives:  a. Clarify age of included participants - ≥ 18 years under objectives but later in inclusion/exclusion criteria states "above the age of 18 years". b. Objective 2c – will time since vaccination be considered e.g. vaccinated >14 days prior to infection? c. Objective 3 – definition of clinically significant disease? 2. Assumptions pg 9 – "The annual incidence of SARS-CoV-2 infection detected by the salivary PCR in those unvaccinated and have no detectable antibodies (unexposed) at baseline will be 12%." Should this not be prevalence instead of incidence as a proportion and not a rate is being given? Similarly, for the other assumptions. 3. Key definitions – include the definition for reinfection - >90 days after previous infection? 4. Establishment of the cohort – it's not clear how the 1200 participants will be selected from the initial 2000 recruited for the survey. Please provide details on this. Figure 1 shows the 4 groups – will there be a specific number of participants in each group? Is there any consideration of age groups when selecting the participants for the cohort? 5. Weekly sample collection: who will transport the samples to the lab each week? The research assistants? 6. The dates of the study should be included in the manuscript. These are included in Figure 1 but not in the main body of the manuscript. 7. How will vaccine status be obtained?
--	---

VERSION 1 – AUTHOR RESPONSE

Reviewer 1: We thank the reviewer for the constructive comments.

Reviewer 2:

We thank the reviewer for the constructive comments. Response to the queries follows

1. First, as far as I understand, the protocol already started mid-July 2021 in India, just after the wave of Delta VOC, where there was a very low number of cases (including primary infection). The reality of the situation is that the current Omicron Wave with BA2 achieved a number of substantial reinfection and already answered to that question.

Response: We agree with the reviewer. This cohort was established well before the Omicron wave. Since the study was a large-scale, prospective public-health study, we wanted to publish the hypothesis, rationale and methodology of the study as a protocol.

We had submitted it first to BMC PH (Submission ID: PUBH-D-21-02047)

well before the Omicron wave. This is our second submission. The delay in publication of the protocol was due to the previous submission and the peer review process. As suggested by the reviewer, we will publish the study results once the period of intensive follow-up is over.

2. Second, my major concerns is that the article does not use the word VOC throughout all the manuscript, while in my opinion this is the main criteria of interest when studying the risk of reinfection.

Reason: We agree and modified the manuscript to include the term Variant of concern. Although we had not mentioned the term in the earlier version, as the reviewer has pointed out, the main objective of including sequencing of all the COVID-19 positive samples was to track and detect the VOC. Line no: 131, pg no: 6.

3. Moreover, the references are old and some new publications are lacking, therefore some statement

such as line 123 "it is crucial to understand the duration of protective immunity" is outdated and referring to 2020, just after you say they do need a one-year hindsight to analyse the risk of reinfection. Thus you might need recent publication because some studied that point.

Reason: The references have been edited to include more recent publications. Line no: 432, 434, 449, pg no: 19.

4. Third, the discussion is lacking the interest of those results, for instance whether or not immunity and risk of further reinfection does protect against severe form of covid and re-admission etc; and whether or not VOC does play a role.

Response: The discussion has been edited to include the aspects suggested by the reviewer. Line no: 404-411, pg no: 18.

5. I'm sorry but, if the study has been achieved then I would rather suggest to publish the findings directly.

Response: As suggested by the reviewer, we will publish the study results once the intensive follow-up period is over. Although the study is at its half-way mark, having the study methodology published as a protocol is valuable to other researchers considering community based cohorts for future variants of concern.

Reviewer 3:

We thank the reviewer for the constructive comments. Response to the queries follows

1. When is the anticipated date of study?

Response: We have included a sub-section on Timeline in the Methods section. Line no: 172-174, pg no: 9.

2. What is the consideration of excluding pregnant women and immunocompromised individuals in serosurvey?

Response: One of the study objectives was to track the cellular and humoral immune correlates of COVID-19 infection, re-infection, and clinically significant disease. Hence, we excluded subjects with pre-existing immunodeficiency states.

Culturally in this region of India, we have a practice where a woman who is pregnant goes to her parents' house for the last few months of pregnancy. The pregnant woman then stays at her parents' home for few months after the delivery of the baby. A follow-up of two years would therefore have much missing data. In addition, pregnancy is a complex immunological state with changes in the cellular and humoral immune responses during various stages of pregnancy. The immune responses may be dysregulated in some, leading to preterm birth and other pregnancy-related complications. Interpretation of the immunological parameters in the light of COVID infection would be difficult. For the above reasons, pregnant women were excluded.

3. How the researcher select one family member in the household? There should be a strict criteria regarding permanent residents, at least already residing in the community for three month etc.

Response: Families who have resided in the area continuously for more than two years were considered as permanent residents and were included.

4. Since this is a two-year prospective cohort, how the research accommodate booster vaccination into the survey?

Response: Information about the booster vaccination status among the participants will be collected during the 6 monthly serology visits. Line no: 256, pg no: 12

5. It seems that there are 4 groups of participants in this cohort, Are there any articles or evidence that support the percentage of infection in these groups as assumptions for sample size estimation? How is the true situation of entire India prior to this study? Statement number 3, annual incidence in unvaccinated and unexposed was estimated at 12% whereas in vaccinated and unexposed was 6%. A study in US demonstrates that incidence rate of unvaccinated was 3-4 times higher. Is it too low to say that the vaccine efficacy is only 50%?

Response: The assumptions were made based on the early findings of the Com-CoV study as there were no published data regarding vaccine efficacy and re-infections prior to the start of CORES study. There are very few data from the community in India, with most data deriving from hospital-based studies. Line no: 197- 199, pg no: 10.

6. Line 255, supposed that symptoms suggesting COVID-19 occurs less than 90 days and tested positive in RT-PCR, will these case counted as reinfection? It is important to clearly define the term reinfection, and repositivity as these terms are different.

Response: We have included the definition of Repositivity as suggested. Line no: 230, pg no: 11.

7. Line 292, is it possible to have other target gene such as ORF1b or S gene in RT-PCR as well?

Response: We have already started the study using targets of S, N gene and RdRp gene. For consistency, we would want to continue with the same targets. Line no: 322, pg no: 15.

8. What is the level of detection of the proposed machine (LOD) and how sensitive the machine in treating saliva sample?

The limit of detection is 100 copies/ ml and the sensitivity in saliva samples is 94% compared to the NP swab. This has been added in the main manuscript. Line no: 323- 325, pg no: 15.

9. If feasible, will the researcher also collected the data regarding COVID medication received by the participants if tested positive? This could answer some questions of whether antivirus or other COVID medications could prevent re-infection.

Response: The study team is collecting the case report form from people who are infected with COVID-19 and these details will be collected as a part of it. However, given the size of the study, we may or may not have sufficient numbers for a meaningful analysis.

10. As for defining seropositive status, kindly cite the minimum level of IgG that could be detected as seropositive according to the selected modalities

Response: The cut off for seropositivity is more than or equal to 33.8 BAU/ml using the automated platform. Line no: 216- 217, pg no: 10.

11. It is important to provide detailed statistical analysis plan, although the plan to use Prentice, Williams and Peterson models is appropriate for recurrent event.

Response: The statistical section has been edited as suggested in the manuscript. Line no: 353- 357, pg no: 16.

Reviewer 4:

We thank the reviewer for her constructive comments.

Response to the queries follows:

1. Clarify age of included participants - ≥ 18 years under objectives but later in inclusion/exclusion criteria states "above the age of 18 years".

Response: The ambiguity has been removed in the manuscript and we have corrected to include age group ≥ 18 years. Line no: 181, pg no: 9.

2. Objective 2c – Will time since vaccination be considered e.g. vaccinated > 14 days prior to infection?

Response: vaccination > 14 days prior to infection will be considered and we have edited in the objectives section to include this statement. line no: 155, pg no: 8.

3. Objective 3 – Definition of clinically significant disease?

Response: We have included the definition of clinically significant disease as suggested. Line no: 228- 229, pg no: 11.

4. Assumptions pg 9- "The annual incidence of SARS-CoV-2 infection detected by the salivary PCR in those unvaccinated and have no detectable antibodies (unexposed) at baseline will be 12%." Should this not be prevalence instead of incidence as a proportion and not a rate is being given? Similarly, for the other assumptions.

Response: The assumptions had been made to estimate the number of cases of infection and re-infection that we expect in the cohort who will be followed for 2 years. Hence, the rate was used instead of a proportion.

5. Key definitions – include the definition for reinfection - > 90 days after previous infection?

Response: The definition of reinfection is included as suggested. Line no: 232-233, pg n o: 11.

6. Establishment of the cohort – it's not clear how the 1200 participants will be selected from the initial 2000 recruited for the survey. Please provide details on this. Figure 1 shows the 4 groups – will there be a specific number of participants in each group? Is there any consideration of age groups when selecting the participants for the cohort?

Response: There are no specific considerations to include participants into the four groups. As they were enrolled, they were categorised into four groups based on their serology and vaccination status. Line no: 255-259, pg no: 12.

7. Weekly sample collection: who will transport the samples to the lab each week? The research assistants?

Response: The Field Research Assistants will visit the participants house, collect and transport the saliva samples to the lab each week details of which has been mentioned in the methodology section of the manuscript. Line no: 265- 266, Pg no: 12.

8. The dates of the study should be included in the manuscript. These are included in Figure 1 but not in the main body of the manuscript.

Response: The dates of the study is included in the manuscript as suggested. Line no:172-174, pg no; 9.

9. How will vaccine status be obtained?

Response: The vaccine status will be obtained during the 6 monthly serology. Line no: 256- 257, pg

no:12.

VERSION 2 – REVIEW

REVIEWER	Herman, Bumi Chulalongkorn University, Public Health
REVIEW RETURNED	03-May-2022

GENERAL COMMENTS	The authors addressed all main issues accordingly. Since the data collection has been conducted, I am looking forward to the initial findings of the study. Thank you
---

REVIEWER	Groome, Michelle National Institute for Communicable Diseases
REVIEW RETURNED	09-May-2022

GENERAL COMMENTS	The authors have adequately addressed my concerns and comments.
---